# Secondary distribution of HIV self-test kits from males to their female sexual partners in two fishing communities in rural Uganda

**Joseph K. B. Matovu**[1,2]*, **Linda Kemigisha**[2], **Geoffrey Taasi**[3], **Joshua Musinguzi**[3], **Rhoda K. Wanyenze**[2], **David Serwadda**[2]

**1** Busitema University Faculty of Health Sciences, Mbale, Uganda, **2** Makerere University School of Public Health, Kampala, Uganda, **3** Ministry of Health, Kampala, Uganda

* jmatovu@musph.ac.ug

**Data Availability Statement:** The dataset used for the analysis has been supplied as S1 Dataset.

## Abstract

Secondary distribution of HIV self-test kits from females to their male partners has increased HIV testing rates in men but little evidence exists on the potential for HIV self-test kits distribution from males to their female partners. We assessed the acceptability of secondary HIV self-test kits distribution from males to their female sexual partners in a fishing community context. This secondary analysis used data from the PEer-led HIV Self-Testing intervention for MEN (PEST4MEN), a pilot interventional study in Buvuma and Kalangala districts in Uganda. At the baseline visit, in July 2022, data were collected from 400 men aged 15+ years who self-reported a HIV-negative or unknown HIV status. Enrolled men were asked to pick two oral fluid HIV self-test kits from a trained male distributor. At the first follow-up visit, in September 2022, men were asked about the number of kits that they received and if they gave kits to anyone, including to their female sexual partners. We used a modified Poisson regression model to determine the factors independently associated with giving kits to sexual partners. Data were analyzed using STATA version 16.0. Of 361 men interviewed at follow-up, 98.3% (355) received at least one kit; 79.7% (283) received two kits. Of those who received two kits, 64% (181) gave the second kit to anyone else; of these, 74.6% (132/177) gave it to a sexual partner. Being currently married (adjusted prevalence ratio [adj. PR] = 1.39; 95% confidence interval [95%CI]: 1.10, 1.75) and having difficulty in reading text prepared in the local language (adj. PR = 1.26; 95%CI: 1.03, 1.55) were significantly associated with men giving kits to their female sexual partners. Ninety-seven per cent (112/132) of the men reported that they knew their sexual partners' HIV self-test results. Of these, 93.7% (n = 105) reported that their partners were HIV-negative while 6.3% (n = 7) reported that they were HIV-positive. Only 28.6% (n = 2) of the HIV-positive sexual partners were reported to have initiated HIV care. Secondary distribution of HIV self-test kits from males to their female sexual partners is well accepted by women in the fishing communities, suggesting that distribution of kits through men in the fishing communities can help to improve HIV testing uptake among their female sexual partners.

**Funding:** This study is part of the EDCTP2 programme that is supported by the European Union (under grant number TMA2019CDF-2729-PEST4MEN, PI: Joseph KB Matovu). The funders had no role in the study design, data collection and analysis, decision to publish, or preparation of the manuscript.

**Competing interests:** The authors have declared that no competing interests exist.

## Introduction

Secondary distribution of HIV self-test kits, a process through which HIV self-test kits are delivered to potential users through someone else, has been used to increase HIV testing uptake among men and other unreachable populations (e.g. men who have sex with men, female sex workers) in most settings [1]. In sub-Saharan Africa, the most common form of secondary HIV self-test kits distribution has been from women to their male partners [2–4]. For instance, evidence shows that pregnant women can successfully deliver HIV self-test kits to their male partners [5–9], and that women can use a variety of strategies to achieve this purpose [10–12]. Studies show that secondary HIV self-test kits distribution from pregnant women to their male partners almost doubled HIV testing uptake among men in Uganda and Kenya [5–8]. There is also growing evidence that social network-based secondary distribution of HIV self-test kits from men to fellow men is well accepted among men in the general population [13, 14] as well as among key populations, including men who have sex with men [15–17]. These findings indicate that secondary distribution is an acceptable approach for reaching men and other unreachable populations with HIV self-testing services.

However, while there is growing evidence of the acceptability of secondary distribution of HIV self-test kits from women to their male partners, there are virtually no studies that have assessed the secondary distribution of HIV self-test kits from men to their female partners. In societies where men make key decisions in the relationship, including decisions on health matters, such as in sub-Saharan Africa [18, 19], understanding whether or not men would accept to take HIV self-test kits to their female partners would be essential in addressing gender norms that normally affect women's decision-making autonomy regarding when, where and how to access health services [20, 21]. Besides, since men tend to fear to test for HIV with their female partners [22], encouraging them to take HIV self-test kits to their female partners could encourage them to test with them, thereby promoting couples' HIV testing uptake in steady relationships. However, it is possible that women may fear to accept HIV self-test kits delivered to them by their male partners for fear of being blamed for bringing HIV into the relationship in the event that they (the female sexual partners) turn out to be HIV-positive [23, 24]. Thus, studies are urgently needed to assess if women would accept to receive and use HIV self-test kits delivered by their male sexual partners.

Studies show that women in the fishing communities tend to be highly mobile [25, 26], moving from one place to the other due to work-related purposes. Because of their high mobility, women may not only be missed through conventional HIV services but they may also have no time to attend facility-based HIV testing services. Such women would benefit from HIV self-testing services when HIV self-test kits have been delivered to them by their male partners. Our study aimed to assess the acceptability of secondary distribution of HIV self-test kits from males to their female sexual partners in two high HIV prevalence island districts in rural Uganda.

## Materials and methods

### Study design, site and population

This secondary analysis used data collected as part of an ongoing, pilot HIV self-testing interventional study that is being implemented in two fishing communities (one in each district) in two high HIV prevalence island districts (Buvuma and Kalangala) located in the Lake Victoria basin in Uganda. Estimates put HIV prevalence in Buvuma at 14% [27] and in Kalangala at 18.8% [28], much higher than the national average of 5.5% among adults aged 15–49 years [29]. The primary objective of the pilot study was to improve HIV testing uptake and linkage

to HIV care among men in the fishing communities through a peer-led HIV self-testing intervention for men (PEST4MEN). The PEST4MEN study methodology has been described previously (Matovu et al. *in press*). In brief, the intervention aimed to improve HIV testing among men in the fishing communities through giving HIV self-test kits to trained male distributors in existing social networks (hereafter referred to as "peer-leaders") to distribute to fellow men within their own social networks. Peer-leaders received three days of training on HIV self-testing processes, basic counseling skills, and how to refer HIV-positive social network members to existing healthcare systems. At the end of the training, each peer-leader was asked to nominate up to 20 men who were known to them and recommend them to the study team for eligibility screening. There were 22 peer-leaders trained in the two districts with a total of 475 men recommended for screening. Of these, 400 men were found to be eligible for study enrolment. Eligible men had to self-report a HIV-negative result or unknown HIV status and should not have tested for HIV at least three months from the time of enrolment. We used a cut-off of three months for any previous HIV testing in line with the Ministry of Health's recommendation for repeat HIV testing among key and priority populations. Enrolling people who last tested within three months would have yielded men who are recent testers, yet the focus of the study was to reach men with infrequent HIV testing behaviors who would be more likely to benefit from HIV self-testing. Eligible men were administered a baseline questionnaire and asked to go to the peer-leader who nominated them to pick two oral fluid HIV self-test kits, one for themselves and the other for someone else, including their female sexual partners.

## HIV self-test kits distribution

Once all social network members recommended by a given peer-leader were screened and those found eligible administered a baseline interview, the study team generated a list of all those who had been interviewed and passed it on to the nominating peer-leader along with two oral HIV self-test kits for each interviewed social network member. Upon receiving the list, the peer-leader was requested to contact their social network members to agree on when and where they would receive their kits. Each social network member was supposed to receive two kits from their peer-leader as per study protocol. The peer-leader was mandated to sensitize their social network members about how to conduct the HIV self-testing exercise and how to read and interpret their results. Social network members were informed that the second kit can be given to anyone within their own network, including their female sexual partner. It was up to them to decide who to give the second kit to. In turn, the social network member would sensitize anyone else that they gave the second kit to.

## Data collection procedures and methods

Data for the pilot study were collected from 400 men at baseline (July 2022) and 361 at the first follow-up visit (September 2022). All interviews were conducted in Luganda, the prominent language spoken by the residents in the two fishing communities. Baseline data were collected on socio-demographic and behavioral characteristics, including HIV testing history and sexual behaviors and willingness to use HIV self-test kits if they were availed to them free of charge, among other characteristics. At the first follow-up visit, men were asked whether or not they had given out the second kit to anyone else, and if they did, we asked them about who they gave the second kit to, whether or not that person accepted to take the kit, and whether or not they think that the person they gave the second kit to actually used it to self-test for HIV. Men who reported that the person that they gave the kit to used it to self-test for HIV were asked if they knew that person's HIV status, and if HIV-positive, if they knew whether or not that person linked to HIV care. No attempt was made to interview the female partners who received

the second kit; data were collected from their male partners who were enrolled into the study. Data were collected by a team of six Research Assistants using questionnaires configured on KoboCollect-enabled mobile phones.

## Measures

The primary outcome was reported acceptability of HIV self-test kits by female sexual partners, defined as female sexual partners accepting to take kits given to them by their male partners and use them to self-test for HIV. The secondary outcome was linkage to HIV care among HIV-positive female sexual partners as assessed from the male partner interviews.

## Data analysis

We conducted descriptive statistics to compute the percentage of men that received two HIV self-test kits and, of these, the percentage of men who gave the second kit to anyone else. Of those that gave the kits to anyone else, we determined the proportion of men who reported that they gave the second kit to their female sexual partners. We then determined the proportion of men that reported if the female sexual partner accepted to take the kit, and what percentage of men reported that their female sexual partners used the kits to self-test for HIV. Among men who reported that their female sexual partners self-tested for HIV, we determined the proportion of those who reported that their female sexual partner self-reported their HIV self-test results to them. Of those that reported an HIV-positive female sexual partner, we asked if the female partner had linked to HIV care. Comparisons between proportions were made using Pearson's Chi-square tests. We used a modified Poisson regression model to determine the factors associated with men giving the second kit to their female sexual partners (as opposed to giving it to other people). All factors with a p-value <0.2 at the bivariate analysis (marital status, number of sexual partners, and men's ability to read text prepared in the local language) and suspected founders (age-group, education, occupation, and study community) were entered into the multivariable analysis. A p-value of <0.05 was considered significant at the multivariable level. Data analysis was conducted using STATA statistical software, version 16.0.

## Ethical considerations

This study was conducted according to the principles expressed in the Declaration of Helsinki. The protocol was approved by the School of Public Health Research and Ethics Committee (Protocol No.: SPH-2021-158) and cleared by the Uganda National Council for Science and Technology (HS2034ES), as per national research guidelines for research among human subjects. All participants provided written informed consent prior to participating in the interviews.

## Results

Table 1 shows the baseline characteristics of the 400 men enrolled into the study from the two fishing communities. Of these, 211 (52.7%) were from Buvuma while 189 (47.2%) were from Kalangala district. Overall, 272 (68.0%) were aged 15–34 years, 257 (64.2%) had primary education as their highest level of education, while 227 (56.7%) were engaged in fishing or fishing-related activities. There were more men in Buvuma who were engaged in fishing or fishing-related activities than in Kalangala district (n = 129 [61.1%] *vs*. n = 98 [51.8%], *P* = 0.06). Nearly 233 (60.0%) of the men were currently married, with a higher proportion of men in Buvuma reporting that they were currently married than those in Kalangala district (n = 146

**Table 1. Baseline socio-demographic characteristics of men enrolled into the study from the two fishing communities, overall and by study district.**

| Characteristic | Kalangala (N = 189, n(%)) | Buvuma (N = 211, n(%)) | Total (N = 400, n(%)) |
|---|---|---|---|
| **Age-group** | | | |
| 18–24 years | 59 (31.2) | 66 (31.3) | 125 (31.2) |
| 25–34 years | 68 (36.0) | 79 (37.4) | 147 (36.7) |
| 35–44 years | 44 (23.3) | 50 (23.7) | 94 (23.5) |
| 45+ years | 18 (9.5) | 16 (7.6) | 34 (8.5) |
| **Highest level of education attained** | | | |
| No education | 10 (5.3) | 19 (9.0) | 29 (7.2) |
| Primary education | 125 (66.1) | 132 (62.6) | 257 (64.2) |
| Post-primary | 54 (28.6) | 60 (28.4) | 114 (28.5) |
| **Marital status** | | | |
| Never married/not in any relationship | 13 (6.9) | 13 (6.2) | 26 (6.5) |
| Never married but in a relationship | 44 (23.3) | 37 (17.5) | 81 (20.3) |
| Currently married | 87 (46.0) | 146 (69.2) | 233 (58.2) |
| Ever married, not in a relationship | 28 (14.8) | 11 (5.2) | 39 (9.7) |
| Ever married, in a relationship | 17 (9.0) | 4 (1.9) | 21 (5.2) |
| **Occupation** | | | |
| Fishing | 65 (34.4) | 88 (41.7) | 153 (38.2) |
| Fishing-related activity | 33 (17.5) | 41 (19.4) | 74 (18.5) |
| Business/commercial | 39 (20.6) | 34 (16.1) | 73 (18.2) |
| Other occupation | 52 (27.5) | 48 (22.7) | 100 (25.0) |
| **Ever tested for HIV** | | | |
| Yes | 168 (88.9) | 160 (75.8) | 328 (82.2) |
| No | 21 (11.1) | 51 (24.2) | 71 (17.8) |

[69.2%] *vs.* n = 87 [46.0%], *P*<0.0001). Considering those who were currently married and other men who were in a sexual relationship but not married, up to 335 (83.7%) can be said to have been in a sexual relationship with a female sexual partner. Three hundred twenty-eight (82.0%) of the men had ever tested for HIV; higher in Kalangala (n = 168, 88.9%) than in Buvuma (n = 60, 75.8%).

Table 2 shows the number and proportion of men that received HIV self-test kits from their peer-leaders in Kalangala and Buvuma districts, as interviewed at the first follow-up visit. Of the 400 men interviewed at baseline, 361 (90.2%) were interviewed at the first follow-up visit. Of these, 355 (98.3%) reported that they received HIV self-test kits from their peer-leaders. Nearly all men (n = 353, 99.4%) reported that they received the kits from a peer-leader who belongs to the same social network that they belonged to. When asked how many HIV self-test kits that they received from their peer-leaders, 283 (79.7%) reported that they received two HIV self-test kits; 70 (19.7%) received only one kit while 2 (0.6%) reported that they received three or more kits. Men in Buvuma were significantly less likely to receive two HIV self-test kits than those in Kalangala (n = 135 [74.6%] *vs.* n = 148 [85.1%], *P* = 0.002). Approximately 312 (87.9%) reported that they felt comfortable or very comfortable to receive HIV self-test kits from the peer-leader who gave them the kits.

Table 3 shows the number and proportion of men that distributed the second kit to someone else. There were 283 men who reported that they received two HIV self-test kits from their peer-leaders. Nearly all of them (n = 269, 95.0%) reported that they think it would be a good thing for men to take HIV self-test kits to their female sexual partners. When asked if they

**Table 2. Receipt of HIV self-test kits from peer-leaders by men living in the fishing communities of Kalangala and Buvuma districts.**

| Variable | Kalangala | Buvuma | Total |
|---|---|---|---|
| | (n/N, %) | (n/N, %) | (n/N, %) |
| **No. of men interviewed at baseline** | **N = 189** | **N = 211** | **N = 400** |
| Number (%) of men interviewed at follow-up | 175 (92.6) | 186 (88.1) | 361 (90.2) |
| Number (%) of men interviewed at follow-up who received kits from their peer-leaders | 174 (99.4) | 181 (97.3) | 355 (98.3) |
| **Peer-leader who gave you the kits belongs to the same social network group that you consider to be your primary social network group** | | | |
| | N = 174 | N = 181 | N = 355 |
| Yes | 172 (98.8) | 181(100) | 353 (99.4) |
| No | 2 (1.1) | 0 (0.0) | 2 (0.6) |
| **Number of kits that men received from their peer-leaders** | | | |
| | N = 174 | N = 181 | N = 355 |
| 1 kit | 25 (14.4) | 45 (24.9) | 70 (19.7) |
| 2 kits | 148 (85.1) | 135 (74.6) | 283 (79.7) |
| 3+ kits | 1 (0.6) | 1 (0.5) | 2 (0.1) |
| **How comfortable was it for you to receive HIV self-test kits from this person?** | | | |
| | N = 174 | N = 181 | N = 355 |
| Comfortable | 104 (59.8) | 71 (39.2) | 175 (49.3) |
| Very comfortable | 49 (28.2) | 88 (48.6) | 137 (38.6) |
| Uncomfortable | 10 (5.7) | 7 (3.9) | 17 (4.8) |
| Very uncomfortable | 11 (6.3) | 13 (7.8) | 24 (6.8) |
| Not sure | 0 (0.0) | 2 (1.1) | 2 (0.6) |

**Table 3. Distribution of the second kit by men living in the fishing communities to other people within their social networks in Kalangala and Buvuma districts.**

| Variable | Kalangala | Buvuma | Total |
|---|---|---|---|
| | (n/N, %) | (n/N, %) | (n/N, %) |
| **Do you think it would be a good thing for men to take HIV self-test kits to their female sexual partners?** | | | |
| | N = 148 | N = 135 | N = 283 |
| Yes | 136 (91.9) | 133 (98.5) | 269 (95.0) |
| No | 6 (4.0) | 2 (1.5) | 8 (3.0) |
| Don't know/not sure | 6 (4.0) | 1 (0.7) | 7 (2.0) |
| **Did you give out the second kit to anyone to use for HIV self-testing?** | | | |
| | N = 148 | N = 135 | N = 283 |
| Yes | 87 (58.8) | 94 (69.6) | 181 (64.0) |
| No | 61 (41.2) | 41 (30.4) | 102 (36.0) |
| **Did the person that you gave the second kit accept to take it?** | | | |
| | N = 87 | N = 94 | N = 181 |
| Yes | 85 (97.7) | 92 (97.9) | 177 (97.8) |
| No | 2 (2.3) | 2 (2.1) | 4 (2.2) |
| **Did you find it easy or difficult to give out the second kit to whoever you gave it to?** | | | |
| | N = 60 | N = 85 | N = 145 |
| Very easy | 38 (63.3) | 61 (71.8) | 99 (68.3) |
| Easy | 21 (35.0) | 23 (27.2) | 44 (30.3) |
| Difficult | 1 (1.7) | 1 (1.2) | 2 (1.4) |
| Very difficult | 0 (0.0) | 0 (0.0) | 0 (0.0) |

gave the second kit to anyone, 181 men (64.0%) responded in the affirmative, with more men in Buvuma (n = 94, 69.6%) reporting that they gave the second kit to anyone than men in Kalangala (n = 87, 58.8%). Men reported that nearly all those that they gave the kits to (n = 177, 97.8%) actually accepted to take them. When asked about how easy or difficult it was for them to give out the second kit to those that they gave the kits to, 143 (98.6%) reported that it was easy or very easy to do so.

Table 4 shows reported HIV self-testing uptake and linkage to HIV care among female sexual partners of the men enrolled into the study. Of the 177 people who reportedly accepted to take the kits, 132 (74.6%) were female sexual partners of the male respondents. One hundred fifteen men (87%) reported that their female sexual partners used the kits to self-test for HIV. When asked if they tested together with their partners, 85 (73.9%) of the men responded in the affirmative, with a higher proportion of men in Buvuma than in Kalangala (n = 54 [78.3%] *vs*. n = 31 [67.4%]; *P* = 0.193) reporting that they self-tested together with their female partners. One hundred twelve men (97%) reported that they knew their female sexual partners' HIV

**Table 4. HIV self-testing uptake and linkage to HIV care among female sexual partners of the men enrolled into the study, as reported by the men.**

| Variable | Kalangala (n/N, %) | Buvuma (n/N, %) | Total (n/N, %) |
|---|---|---|---|
| **What was your relationship with the person that you gave the second kit to?** | | | |
| | **N = 85** | **N = 92** | **N = 177** |
| Primary/steady partner | 31 (36.5) | 59 (64.1) | 90 (50.8) |
| Casual partner | 2 (2.3) | 3 (3.3) | 5 (2.8) |
| Girlfriend | 26 (30.6) | 11 (12.0) | 37 (20.9) |
| Relative | 4 (4.7) | 4 (4.3) | 8 (4.5) |
| Other person | 22 (25.9) | 15 (16.3) | 37 (20.9) |
| **Proportion of men reporting that their female sexual partners (*primary/steady partner/casual partner/girlfriend*) used the kits to self-test for HIV** | | | |
| | **N = 59** | **N = 73** | **N = 132** |
| Used kit | 46 (77.9) | 69 (94.5) | 115 (87.1) |
| Did not use the kit | 8 (13.6) | 2 (2.7) | 10 (7.6) |
| Don't know | 5 (8.5) | 2 (2.7) | 7 (5.3) |
| **Proportion of men reporting that they self-tested together with their female sexual partners, among those that used the kit** | | | |
| | **N = 46** | **N = 69** | **N = 115** |
| Yes | 31 (67.4) | 54 (78.3) | 85 (73.9) |
| No | 15 (32.6) | 15 (21.7) | 30 (26.1) |
| **Proportion of men reporting that they knew their female sexual partners' HIV self-test results (*either through female partner telling them or testing together with them*)** | | | |
| | **N = 46** | **N = 69** | **N = 115** |
| Yes | 46 (100) | 66 (95.6) | 112 (97.4) |
| No | 0 (0.0) | 3 (4.3) | 3 (2.6) |
| **Proportion of men reporting that the HIV self-test results of their female sexual partners** | | | |
| | **N = 46** | **N = 66** | **N = 112** |
| HIV negative | 45 (97.8) | 60 (90.9) | 105 (93.7) |
| HIV positive | 1 (2.2) | 6 (9.1) | 7 (6.3) |
| **Proportion of men reporting that their HIV female sexual partners linked to HIV care** | | | |
| | **N = 1** | **N = 6** | **N = 7** |
| Yes | 0 (0.0) | 2 (33.3) | 2 (28.6) |
| No | 1 (100) | 3 (50.0) | 4 (57.1) |
| Don't know | 0 (0.0) | 1 (16.7) | 1 (14.3) |

**Table 5. Factors associated with men living in the fishing communities giving the second kit to a female sexual partner.**

| Characteristic | Percentage of men that gave the second kit to a female sexual partner (n/N, %) | Crude prevalence ratio and 95% confidence interval [95%CI] | p-value | Adjusted prevalence ratio [adj. PR] and 95%CI | p-value |
|---|---|---|---|---|---|
| **Age-group** | | | | | |
| 18–24 years | 48/64 (75.0) | 1.00 | - | 1.00 | - |
| 25–34 years | 51/65 (78.5) | 1.05 (0.86, 1.27) | 0.64 | 0.89 (0.74, 1.07) | 0.21 |
| 35–44 years | 26/36 (72.2) | 0.96 (0.75, 1.23) | 0.76 | 0.83 (0.67, 1.03) | 0.09 |
| 45+ years | 7/12 (58.3) | 0.78 (0.47, 1.28) | 0.32 | 0.72 (0.46, 1.12) | 0.15 |
| **Highest level of education attained** | | | | | |
| No education | 7/11 (63.6) | 1.00 | - | 1.00 | - |
| Primary level | 75/105 (71.4) | 1.12 (0.70, 1.78) | 0.63 | 0.83 (0.54, 1.26) | 0.39 |
| Post-primary level | 50/61 (82.0) | 1.29 (0.81, 2.05) | 0.28 | 0.86 (0.56, 1.30) | 0.47 |
| **Marital status** | | | | | |
| Never married/single | 34/50 (68.0) | 1.00 | - | 1.00 | - |
| Ever married | 6/21 (28.6) | 0.42 (0.21, 0.85) | 0.02 | 0.48 (0.24, 0.96) | 0.04 |
| Currently married | 92/106 (86.8) | 1.28 (1.04, 1.57) | 0.02 | **1.39 (1.10, 1.75)** | 0.01 |
| **Occupation** | | | | | |
| Fishing | 35/47 (74.5) | 1.00 | - | 1.00 | - |
| Fishing-related | 47/61 (77.0) | 1.03 (0.83, 1.28) | 0.76 | 1.01 (0.82, 1.25) | 0.89 |
| Business/commercial | 25/38 (65.8) | 0.88 (0.66, 1.17) | 0.39 | 0.82 (0.65, 1.04) | 0.10 |
| Other occupation | 25/31 (80.6) | 1.08 (0.85, 1.38) | 0.52 | 1.04 (0.85, 1.27) | 0.71 |
| **Study community** | | | | | |
| Kasaali-B | 73/92 (79.3) | 1.00 | - | 1.00 | - |
| Mwena | 59/85 (69.4) | 0.87 (0.73, 1.04) | 0.14 | 0.97 (0.83, 1.14) | 0.73 |
| **Number of sexual partners (*in the past 3 months*)** | | | | | |
| 1 | 69/81 (85.2) | 1.00 | - | 1.00 | - |
| 2+ | 63/96 (65.6) | 0.77 (0.65, 0.91) | <0.01 | 0.91 (0.78, 1.07) | 0.25 |
| **Ability to read text in local language** | | | | | |
| Not able to read at all | 41/62 (66.1) | 1.00 | - | 1.00 | - |
| Reads with difficulty | 41/49 (83.7) | 1.26 (1.02, 1.57) | 0.03 | **1.26 (1.03, 1.55)** | 0.02 |
| Reads with ease | 50/66 (75.7) | 1.14 (0.91, 1.43) | 0.24 | 1.04 (0.88, 1.34) | 0.42 |

self-test results (either through being told by them or through couples' HIV self-testing). Of these, 105 men (93.7%) reported that their female sexual partners were HIV-negative while 7 men (6.3%) reported that their female partners were HIV-positive. Of those reporting that their female partners were HIV-positive, only two men (28.6%) reported that their HIV-positive female sexual partners were linked to HIV care.

Table 5 shows the factors that were independently associated with men's distribution of the second kit to their female sexual partners, among those that received two kits. At the bivariate analysis, being currently married and reporting difficulties in reading text prepared in *Luganda*, the primary local language spoken in the area, were associated with men giving the second kit to their female sexual partners. This association remained strong even after adjusting for potential confounders. Being currently married versus being single or never married (adjusted prevalence ratio [adj. PR] = 1.39; 95% confidence interval [95%CI]: 1.10, 1.75) and having difficulty reading text prepared in the local language versus not being able to read at all (adj. PR = 1.26; 95%CI: 1.03, 1.55) were significantly associated with men giving the second kit to a female sexual partner.

## Discussion

Our study assessed the reported acceptability of secondary distribution of HIV self-test kits from males to their female sexual partners in a fishing community setting. Study findings show that: a) 64% of the men who received two kits gave the second kit to someone else, of these, 75% gave them to their female sexual partners; b) 87% of the men reported that the female sexual partners that they gave the kits to used them to self-test for HIV (of these, 74% reported that they self-tested for HIV together with their female sexual partners), but c) only 29% of the men reported that their HIV-positive female self-testers were linked to HIV care. In general, these findings show that distribution of HIV self-test kits from males to their female sexual partners is acceptable with a high proportion of HIV self-testing uptake but linkage to HIV care remains sub-optimal, requiring innovative approaches that are probably aligned to the women's way of life. Some reports indicate that women in the fishing communities are highly mobile [25, 31]; this is likely to have been the reason why those who were HIV-positive failed to link to HIV care. Thus, any innovative approaches intended to link these women to HIV care should consider their high mobility patterns coupled with the general fisherfolk's reluctance to utilize health facility-based services due to their location far away from the fishing sites [30, 31].

Our finding that only 64% of men that received two kits gave the second kit to someone else suggests that up to 36% of these men, especially in Kalangala (41.2%) than in Buvuma (30.4%), did not give out the second kit to anyone. We did not inquire into the reasons why these men did not give out the second kit to anyone; an area that we intend to inquire into as part of post-intervention qualitative assessment. However, the finding that nearly all people who were given the second kit (98%) reportedly accepted to take them is a clear indication of the potential demand for kits, which could be met with increased distribution through alternative HIV self-test kits distribution channels. It is encouraging that nearly three-quarters of the men gave the kits to their female sexual partners, and that 87% of the men reported that their female sexual partners, who received the kits, used them to self-test for HIV. Besides, 95% of the men reported that it would be a good thing for men to take HIV self-test kits to their female sexual partners. These findings suggest that secondary distribution of HIV self-test kits from males to their female sexual partners can increase HIV testing uptake among female sexual partners of men living in the fishing communities who may miss conventional health facility-based HIV testing services due to their mobility patterns [25].

Although we did not primarily aim to assess the uptake of couples' HIV self-testing as part of the main study, our finding that 74% of the men reportedly tested together with their female sexual partners suggests that the delivery of HIV self-test kits by men to their female partners could have motivated men to self-test together with their female sexual partners Similar results have been reported in a study in Kenya where couples' HIV testing was higher among pregnant women who received HIV self-test kits to take to their male partners than in those who were given cards inviting their male partners to come to the health facility to test for HIV [6]. While the Kenyan study focused on women who delivered kits to their male partners, our study of men who delivered HIV self-test kits to their female partners suggests that secondary distribution of HIV self-test kits can have additional benefits, including couples' HIV testing, given that in most cases women are interested in testing together with their male partners but men tend to be reluctant to do so [22]. Further research is warranted to assess the potential of male partner-delivered HIV self-test kits in improving couples' HIV testing in established relationships within the fishing communities.

We found that reported linkage to HIV care among the HIV-positive females was low, with only 29% of the female sexual partners reported to have linked to HIV care, based on reports

from their male partners. We don't know why there were fewer women living with HIV who reportedly linked to HIV care since we did not interview the women themselves. However, based on evidence from other studies [25, 30], it is likely that HIV-positive women may have faced barriers to linkage to HIV care that are related to their high mobility patterns coupled with residence in remote fishing locations that are far away from the main health facilities [25, 30, 31]. Thus, efforts to improve linkage to HIV care among women living with HIV in the fishing communities may require use of innovative approaches, including community health outreaches to remote fishing locations [32, 33]. Evidence suggests that community-based ART initiation can help to overcome some of the barriers that unreachable populations, such as those living in the fishing communities, often face in linking to HIV care [34]. However, given that current evidence is based on studies conducted outside fishing community settings, further research is needed to understand if community-based health outreaches can help to improve linkage to HIV care among highly mobile women living with HIV in the fishing communities.

Being currently married was significantly associated with men giving out the second kit to female sexual partners. It is likely that the men could have had interest in knowing their female partners' HIV status or wanted to self-test for HIV with them, as noted above. However, this aspect requires further inquiry since we did not assess the reasons why currently married men were more likely to give the second kit to their female sexual partners than those who were single or not married. We also found that men who had difficulty in reading text prepared in the local language were significantly more likely to give the second kit to their female sexual partners than those who were not able to read at all. We can't tell why this was the case since we did not assess the reasons why men with reading difficulties were more likely to give the second kit to their female sexual partners than those who could not read at all. Further inquiry is warranted to fully explain this phenomenon.

This study had a number of limitations and strengths. First and foremost, the study relied on self-reports about the female partners' use of HIV self-test kits to test for HIV. We did not interview the female sexual partners of the men enrolled into the study since this wasn't the primary focus of the study. Besides, we did not specify who was meant to receive the second kit, among those that preferred to give the kit to anyone else. It was at the time of analysis that we realized that three-quarters of the men gave the second kit to their female sexual partners. As such, no study tools were designed to collect data from the female partners of the men enrolled into the study. We don't know if the women willingly accepted to receive the kits or to self-test for HIV. We also don't know if the women were first-time HIV-positive testers, and if so, whether or not they sought confirmatory HIV testing, as recommended. Thus, our study findings, based on men's reports of their female sexual partners' HIV self-testing behaviors, should be interpreted with caution. Future studies should aim to interview female sexual partners of men enrolled into the study in their capacity as secondary HIV self-test kits recipients to document if they willingly accepted to take the kit and use it to self-test for HIV. The other limitation is that we did not exclusively focus on the male fisherfolk per se, which may affect the generalization of our study findings to the male fisherfolk in other fishing community settings. However, we believe that a focus on men in the fishing communities in general rather specifically to the male fisherfolk offers a broader perspective of understanding the HIV testing behaviors of men in general, which is essential to improve HIV testing uptake among men in these settings. Thus, despite the above-mentioned study limitations, our study explored a novel approach for reaching women in the fishing communities with HIV self-testing through their male partners. Our findings indicate a promising HIV self-test kits distribution approach, and, if adopted across settings, secondary distribution of HIV self-test kits from men to their female partners could be the game changer needed to improve HIV testing uptake among

highly mobile women in the fishing communities who tend to test less frequently than their male counterparts [35].

## Conclusion

Our findings show that secondary distribution of HIV self-test kits from males to their female sexual partners is acceptable in a fishing community setting. These findings suggest that distribution of HIV self-test kits through men living in the fishing communities can help to reach their female sexual partners who are equally mobile and may face challenges accessing conventional HIV services due to their high mobility patterns.

## Supporting information

**S1 Checklist. STROBE checklist.**
(DOC)

**S1 Dataset. Dataset used in the analysis of data.**
(DTA)

## Acknowledgments

We acknowledge the support of the district and community leadership in supporting the implementation of the study; the peer-leaders in Kalangala and Buvuma for distributing the HIV self-test kits to men within their social networks; research assistants for collecting the data, and the study participants for their time in participating in the study.

## Author Contributions

**Conceptualization:** Joseph K. B. Matovu, Rhoda K. Wanyenze, David Serwadda.

**Data curation:** Linda Kemigisha.

**Formal analysis:** Joseph K. B. Matovu, Linda Kemigisha.

**Funding acquisition:** Joseph K. B. Matovu.

**Methodology:** Joseph K. B. Matovu, Geoffrey Taasi, Joshua Musinguzi, Rhoda K. Wanyenze, David Serwadda.

**Project administration:** Joseph K. B. Matovu.

**Resources:** Geoffrey Taasi, Joshua Musinguzi.

**Supervision:** Joseph K. B. Matovu.

**Validation:** Joseph K. B. Matovu, Linda Kemigisha, Geoffrey Taasi, Joshua Musinguzi, Rhoda K. Wanyenze, David Serwadda.

**Writing – original draft:** Joseph K. B. Matovu.

**Writing – review & editing:** Joseph K. B. Matovu, Linda Kemigisha, Geoffrey Taasi, Joshua Musinguzi, Rhoda K. Wanyenze, David Serwadda.

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
