## [Decision Letter · Decision Letter 0]

17 Aug 2023

PGPH-D-23-01082

Secondary distribution of HIV self-test kits from male fisherfolk to their female sexual partners in two fishing communities in rural Uganda

Dear Dr. Matovu,

Thank you for submitting your manuscript to PLOS Global Public Health. After careful consideration, we feel that it has merit but does not fully meet PLOS Global Public Health’s publication criteria as it currently stands. Therefore, we invite you to submit a revised version of the manuscript that addresses the points raised during the review process.

EDITORS COMMENTS:

The main objective of this manuscript is to assess the acceptability of Secondary distribution of HIV self-test kits from male fisherfolk to their female sexual partners, however, it is noted that nearly 40% of the men participating in the study are not involved in fishing or fishing related occupations- please clarify or alternatively revise the title and objectives accordingly.

While you sought to assess the acceptability of secondary distribution of HIV self test kits- the acceptability is self reported by the participants (you have mentioned this in your limitations)- the tests may have been forced on the partners or falsely reported- there is therefore no true evidence of the acceptability of the tests- and, therefore, may be termed 'reported acceptability.'

How was the sample size of 400 participants decided?

Were the participants taught how to use the test?  Were they told to communicate how to use the testing kit to those they gave it to?- please include this in the methods.

Results- Actual numbers to be mentioned with percentages in brackets.

Given that the characteristics of the partners who accepted the tests are not available, predictors of having shared the test or both partners having done the test may be presented. 

Kindly avoid assumptions such as for example: 'kept the second test for their own use or sold it for money' without sufficient evidence.

Limitations need to include why the women who received the tests were not included. Likely biases such as selection bias (i.e., regarding those who received the kits) need to be discussed. How likely is it that the women willingly accepted the test? While this has been mentioned in the need for the study there is no information regarding this subsequently. The data was secondary- a comment on the quality of the data and how that impacts the results is needed.

We look forward to receiving your revised manuscript.

Kind regards,

Rashmi Josephine Rodrigues, M.D., Ph.D.

Academic Editor

Journal Requirements:

1. Please ensure that Funding Information and Financial Disclosure Statement are matched.

2. In the Funding Information you indicated that no funding was received. Please revise the Funding Information field to reflect funding received.

Reviewers' comments:

Reviewer's Responses to Questions

**Comments to the Author**

1. Does this manuscript meet PLOS Global Public Health’s publication criteria? Is the manuscript technically sound, and do the data support the conclusions? The manuscript must describe methodologically and ethically rigorous research with conclusions that are appropriately drawn based on the data presented.

Reviewer #1: Yes

Reviewer #2: Yes

2. Has the statistical analysis been performed appropriately and rigorously?

Reviewer #1: No

Reviewer #2: Yes

3. Have the authors made all data underlying the findings in their manuscript fully available (please refer to the Data Availability Statement at the start of the manuscript PDF file)?

Reviewer #1: Yes

Reviewer #2: Yes

4. Is the manuscript presented in an intelligible fashion and written in standard English?

Reviewer #1: Yes

Reviewer #2: Yes

5. Review Comments to the Author

Reviewer #1: Firstly, I would like to appreciate the immense effort taken by the authors to carry out this study among a vulnerable population, based on occupation as well as gender. This is an all in all well written paper, although I do have the following reservations about it.

1) a flow chart describing the recruitment and follow-up process will be appreciated. details of number of participants recruited, found eligible, reasons and proportions of ineligibility, lost to follow-up and further outcomes.

2) language in which the interview was carried out can be added in the description of data collection

3) how was interviewer bias addressed?

4) Of those who did not distribute the self test kit, what were the reasons?

5) A multivariate logistic regression maybe to tease out those variables that have an impact on your primary outcome (as very few did not take up testing kits, one kit v/s 2 or more kits can be the outcome variable).

6) Scope for mixed methods studies with the women folk who received the test kits can be mentioned in the limitations of the study

Reviewer #2: The paper requires minor correction. Otherwise, the paper meets the PLOS Global Public Health standards and guidelines.

Line 40: 88.0% of 283 men is not 181.

Line 110: Why was this an eligible criterion: "Should not have tested for HIV at least three months from the time of enrolment"

Line 125: Why was no attmept made to interview the female partners who received the second kit (Because the authors did not attempt to interview the female partners who received the second kit, the second part of the definition of acceptance may remain unfulfilled)

Line 158: Why weren't all 400 men engaged in fishing or fishing related activities? More than 40% of the study subjects were involved in non-fishing activities. In that case, the findings of this study cannot be generalized to fisher folk.

Line 232, 233: Many of the statements in the discussion are 'assumptions' made without any corroboration from the findings of this study. There are some statements that are assumptions, but they are backed by reference. But many aren't. Discussion may have to be re-written.

6. PLOS authors have the option to publish the peer review history of their article (what does this mean?). If published, this will include your full peer review and any attached files.

**Do you want your identity to be public for this peer review?** For information about this choice, including consent withdrawal, please see our Privacy Policy.

Reviewer #1: No

Reviewer #2: No

---

## [Decision Letter · Decision Letter 1]

27 Oct 2023

Secondary distribution of HIV self-test kits from males to their female sexual partners in two fishing communities in rural Uganda

PGPH-D-23-01082R1

Dear Dr. Matovu,

We are pleased to inform you that your manuscript 'Secondary distribution of HIV self-test kits from males to their female sexual partners in two fishing communities in rural Uganda' has been provisionally accepted for publication in PLOS Global Public Health.

Best regards,

Rashmi Josephine Rodrigues, M.D., Ph.D.

Academic Editor

Reviewer Comments (if any, and for reference):

Reviewer's Responses to Questions

**Comments to the Author**

1. If the authors have adequately addressed your comments raised in a previous round of review and you feel that this manuscript is now acceptable for publication, you may indicate that here to bypass the “Comments to the Author” section, enter your conflict of interest statement in the “Confidential to Editor” section, and submit your "Accept" recommendation.

Reviewer #1: All comments have been addressed

Reviewer #2: All comments have been addressed

2. Does this manuscript meet PLOS Global Public Health’s publication criteria? Is the manuscript technically sound, and do the data support the conclusions? The manuscript must describe methodologically and ethically rigorous research with conclusions that are appropriately drawn based on the data presented.

Reviewer #1: Yes

Reviewer #2: Yes

3. Has the statistical analysis been performed appropriately and rigorously?

Reviewer #1: Yes

Reviewer #2: Yes

4. Have the authors made all data underlying the findings in their manuscript fully available (please refer to the Data Availability Statement at the start of the manuscript PDF file)?

Reviewer #1: Yes

Reviewer #2: Yes

5. Is the manuscript presented in an intelligible fashion and written in standard English?

Reviewer #1: Yes

Reviewer #2: Yes

6. Review Comments to the Author

Reviewer #1: Thank you for your response. You may want to consider adding a statement on the recruitment details being described in the primary article for reader reference.

Reviewer #2: The paper is now suitable to be published in PLOS Global Public Health. The authors have addressed all queries and have drawn reasonable conclusions from the data. This paper can now be published.

7. PLOS authors have the option to publish the peer review history of their article (what does this mean?). If published, this will include your full peer review and any attached files.

**Do you want your identity to be public for this peer review?** For information about this choice, including consent withdrawal, please see our Privacy Policy.

Reviewer #1: No

Reviewer #2: No
